# Comparative Transcriptome Profiling of CMS-D2 and CMS-D8 Systems Characterizes Fertility Restoration Genes Network in Upland Cotton

**DOI:** 10.3390/ijms241310759

**Published:** 2023-06-28

**Authors:** Xiatong Song, Meng Zhang, Kashif Shahzad, Xuexian Zhang, Liping Guo, Tingxiang Qi, Huini Tang, Hailin Wang, Xiuqin Qiao, Juanjuan Feng, Yang Han, Chaozhu Xing, Jianyong Wu

**Affiliations:** 1Zhengzhou Research Base, National Key Laboratory of Cotton Bio-Breeding and Integrated Utilization, School of Agricultural Sciences, Zhengzhou University, Zhengzhou 450001, China; songxiatong11@163.com; 2National Key Laboratory of Cotton Bio-Breeding and Integrated Utilization, Key Laboratory for Cotton Genetic Improvement, Ministry of Agriculture and Rural Affairs, Institute of Cotton Research of Chinese Academy of Agricultural Sciences, Anyang 455000, China; zhangmeng910305@163.com (M.Z.); kashifshahzad85@yahoo.com (K.S.); zhangxuexian@caas.cn (X.Z.); guolp@cricaas.com.cn (L.G.); qitx@cricaas.com.cn (T.Q.); tanghn@cricaas.com.cn (H.T.); wanghal@126.com (H.W.); qiaoxiuqin@caas.cn (X.Q.); fengjuanjuan-cotton@outlook.com (J.F.); hanqx970226@163.com (Y.H.)

**Keywords:** upland cotton (*Gossypium hirsutum* L.), CMS-D2, CMS-D8, *Rf_1_*, *Rf_2_*, DEGs

## Abstract

Resolving the genetic basis of fertility restoration for cytoplasmic male sterility (CMS) can improve the efficiency of three-line hybrid breeding. However, the genetic determinants of male fertility restoration in cotton are still largely unknown. This study comprehensively compared the full-length transcripts of CMS-D2 and CMS-D8 systems to identify potential genes linked with fertility restorer genes *Rf_1_* or *Rf_2_*. Target comparative analysis revealed a higher percentage of differential genes in each restorer line as compared to their corresponding sterile and maintainer lines. An array of genes with specific expression in the restorer line of CMS-D2 had functional annotations related to floral development and pathway enrichments in various secondary metabolites, while specifically expressed genes in the CMS-D8 restorer line showed functional annotations related to anther development and pathway enrichment in the biosynthesis of secondary metabolites. Further analysis identified potentially key genes located in the target region of fertility restorer genes *Rf_1_* or *Rf_2_.* In particular, *Ghir_D05G032450* can be the candidate gene related to restorer gene *Rf_1_*, and *Ghir_D05G035690* can be the candidate gene associated with restorer gene *Rf_2_*. Further gene expression validation with qRT-PCR confirmed the accuracy of our results. Our findings provide useful insights into decoding the potential regulatory network that retrieves pollen fertility in cotton and will help to further reveal the differences in the genetic basis of fertility restoration for two CMS systems.

## 1. Introduction

Cotton is grown worldwide as one of the most important fiber- and oil-producing crops. Staggered cotton yield potential of genotypes, along with instability in their performance with climate change, is a major limitation of sustainable cotton production [1]. The utilization of heterosis can therefore be an important strategy to reduce the recent decline in cotton productivity. Heterosis has been used to produce superior cultivars in terms of yield, adaptability, disease resistance, and fiber quality [2]. Artificial emasculation and pollination (AEP) and male sterility (MS) systems are commonly applied to produce hybrids in cotton [3]. AEP has been a conventional way of hybrid breeding with time-consuming, costly, and lower seed purity [4]. The CMS is a maternally inherited trait that produces non-functional pollen grains due to impaired cytoplasmic and nuclear interaction [5]. In recent years, the application of CMS has gradually become a research hotspot in the field of cotton heterosis utilization. It avoids the hassle of anther removal and is considered a supreme way of commercial hybrid seed production in cotton [4,6,7,8].

So far, CMS-D2 and CMS-D8 are the two main systems that have already been well-progressed to exploit heterosis to improve cotton yield and quality [4]. The CMS system depends on three main breeding lines: male-sterile (A), restorer (R), and maintainer (B). However, the restorer line is the key component of the CMS system, and its interaction with the A line produces fertile F_1_ hybrids that are superior in yield, fiber quality, and disease resistance [9]. In cotton, pollen fertility is retrieved by the fertility restorer genes *Rf_1_* and *Rf_2_*. It is well known that *Rf_1_* from *G. harknessii* (D2-2 genome) can revive pollen fertility in sterile lines of CMS-D2 and CMS-D8. In contrast, *Rf_2_* from *G. trilobum* (D8) can revive pollen fertility only in the sterile line of CMS-D8 [10,11,12,13,14,15,16,17]. These findings indicated that CMS-D2 and *Rf_1_* have important relationships with CMS-D8 and *Rf_2_*. So, research studies that reveal the precise relationships among CMS-D2 and CMS-D8 are essential to exploring the genetic architecture of sterility/fertility in cotton. Over the past few years, RNA sequencing (RNA-Seq) platforms have developed rapidly [18] and become an indispensable tool to analyze transcripts in several crops, including sweet potatoes [19], alfalfa [19], sugarcane [20], and tea [21]. The transcriptome analyses in cotton revealed key genes related to fiber development [22,23], fungus defense mechanisms [24], and leaf senescence [25], among others. In addition, the molecular mechanisms of fertility traits were widely investigated with short-length transcript analyses in CMS cotton. These recent studies predict that male sterility in CMS-D8 is likely linked to cell wall expansion [26]. While the genes related to cotton circadian rhythm possibly influenced the pollen fertility in CMS-D2 [27]. Another study revealed that the fertility of CMS-D2 could be restored by miRNA-target gene pairs that cleave a pentatricopeptide repeat (PPR) gene [28]. In addition, reactive oxygen species (ROS) may serve as important signaling molecules for cytoplasmic male sterility in CMS-D8 cotton [29].

Apart from these, many studies have been performed on genetic mapping and the development of markers related to *Rf_1_* and *Rf_2_*. Liu et al. [30] reported two random amplifications of polymorphic DNA (RAPD) and three simple sequence repeat (SSR) markers linked but not co-segregated with *Rf_1_*. Then, sequence tag site (STS) tags (UBC1471400, UBC607500, and UBC679700) co-segregated with *Rf_1_* were identified [31]. Yin et al. [32] limited the possible location of the *Rf_1_* gene to a 100-kb region. Other studies verified that SSR marker NAU211 was linked with the *Rf_1_* gene with a genetic distance of 0.163 cM [33], BNL3535 with a close genetic distance of 0.049 cM, and NAU3652 on the other side with a genetic distance of 0.2 cM [34]. In addition, Wu et al. [35] developed four insertion/deletion (InDel) markers (InDel-1891, InDel-3434, InDel-7525, and InDel-9356) tightly linked to the *Rf_1_* gene and fine-mapped *Rf_1_* at a genetic distance below 0.25 cM. The mapping studies on the *Rf_2_* gene revealed that an RAPD marker, UBC188_500_, was the closest marker to the *Rf_2_* gene at a genetic distance of 2.9 cM [36], and two amplified fragment length polymorphism (AFLP) markers and one SSR marker were verified to be linked with the *Rf_2_* gene [37]. Feng et al. [38] further narrowed the candidate interval of *Rf_2_* by single nucleotide polymorphisms (SNP) through high-throughput genotyping and InDel markers. Further developed the InDel-1892 marker to differentiate between *Rf_2_* and *Rf_1_* simultaneously [39]. Previous research reported that *Rf_1_* and *Rf_2_* are not allelic but tightly linked at a genetic distance of 0.93 cM on chromosome D05 [17]. Although the development of genetic makers has facilitated molecular breeding, the cloning of *Rf_1_* and *Rf_2_* genes is still not performed in cotton.

The recently developed full-length transcript sequencing technology has emerged as a substantial source for exploring the genetic landscape of target traits. To date, no systematic study has been performed on the association between the fertility restorer genes *Rf_1_* and *Rf_2_* of CMS-D2 and CMS-D8. This understanding could enhance the existing knowledge of fertility restoration mechanisms. The present study compared the transcriptional datasets of CMS-D2 and CMS-D8 lines using full-length transcript sequencing. Further analyzed gene expression differences, identified fertility-related genes, mapped key genes into target regions of *Rf_1_*/*Rf_2_*, and confirmed the accuracy of the results with real-time quantitative reverse transcription PCR (qRT-PCR). Taken together, our results open a new door for further disclosing the differences in the genetic basis of pollen fertility restoration for both CMS systems in cotton.

## 2. Results

### 2.1. Differentially Expressed Genes (DEGs) Analysis in Both CMS-D2 and CMS-D8 Systems

Our group previously characterized full-length anther transcripts in both CMS-D2 and CMS-D8 systems [40,41]. Here, a comprehensive analysis was performed to analyze the similarities and differences of DEGs among A, B, and R lines between the two cotton CMS systems. The comparisons of gene expression were performed between B vs. R and A vs. R. The comparison of B and R revealed 1692 DEGs in CMS-D2, whereas 2297 DEGs were identified in CMS-D8 (Figure 1a). There were 189 overlapped DEGs between CMS-D2 and CMS-D8, which account for 11.17% and 8.23% of total DEGs, respectively. In the comparison of A and R, 804 DEGs in CMS-D2 and 4112 DEGs in CMS-D8 showed significant expression changes. The 103 DEGs were co-expressed in both CMS-D2 and CMS-D8, which account for 12.81% and 2.50% of the total number of DEGs, respectively (Figure 1a). Further analysis stated that B vs. R had 871 specific DEGs in CMS-D2 and 756 specific DEGs in CMS-D8. In A vs. R, there were 389 and 2250 specific DEGs in CMS-D2 and CMS-D8, respectively (Figure 1b).

The distribution of DEGs revealed that the comparison of B vs. R had 625 up-regulated DEGs in the CMS-D2 system and 829 up-regulated DEGs in the CMS-D8 system (Figure 2a). The number of down-regulated DEGs was 1067 in CMS-D2 and 1468 in CMS-D8. In the comparison of A vs. R, 507 and 2211 DEGs exhibited up-regulation in CMS-D2 and CMS-D8 systems, respectively. There were 297 down-regulated DEGs in CMS-D2 and 1901 down-regulated DEGs in CMS-D8 (Figure 2a). Among up-regulated DEGs, 204 DEGs were up-regulated in the D2 R line compared to the B and A lines. While 557 DEGs were up-regulated in the D8 R line compared to the B and A lines. Moreover, 184 and 505 DEGs specifically showed up-regulation in CMS-D2 and CMS-D8, respectively. Only eight DEGs with up-regulated expression changes overlapped between the CMS-D2 and CMS-D8 systems (Figure 2b). For down-regulated DEGs, 110 DEGs in the D2 R line and 682 DEGs in the D8 R line had down-regulation as compared to the B and A lines. The 100 and 651 DEGs were specific to CMS-D2 and CMS-D8, respectively. However, only five DEGs with down-regulation overlapped in both CMS-D2 and CMS-D8 (Figure 2c). In particular, our comparative analysis determined that the R lines had significant genetic differences from the other two lines. In addition, the higher number of genes specific to each R line suggests the complexity of the fertility restoration mechanism in CMS-D2 and CMS-D8 cotton.

### 2.2. Chromosome Co-Location Analysis of Specifical DEGs of Both CMS-D2 and CMS-D8 R Lines

The chromosome location analysis found that 184 and 505 DEGs specifically up-regulated in the R lines of CMS-D2 and CMS-D8 were distributed on different chromosomes and scaffolds. The 118 DEGs were distributed to the 25 chromosomes of *G. hirsutum* (excluding chromosome D02) and the 66 DEGs on 54 scaffolds in the R line of CMS-D2. In the R line of CMS-D8, 499 specifically up-regulated genes were distributed across 26 chromosomes of *G. hirsutum* and six DEGs on six scaffolds (Figure 3a). The down-regulated DEGs in the R line—100 DEGs of CMS-D2 and 651 DEGs of CMS-D8—were also distributed on different chromosomes and scaffolds. Among the genes specifically down-regulated in the R line, 99 DEGs of CMS-D2 were distributed across the 26 chromosomes of *G. hirsutum* and one DEG on one scaffold. Whereas 647 DEGs of CMS-D8 were distributed across the 26 chromosomes of *G. hirsutum* and four DEGs on four scaffolds (Figure 3b). Importantly, it was observed that Chr_D05 (with restorer genes *Rf_1_* and *Rf_2_*) and its homologous Chr_A05 had more differential genes regulation both in CMS-D2 as well as in CMS-D8. Theoretically, the differential regulation of genes specific to restorer lines and distributed on Chr_D05 most probably contribute to the generation of fertile pollen in cotton.

### 2.3. Gene Ontology (GO) Function Enrichment Analysis of Specific DEGs in the R Line

To understand how DEGs specific to R lines affect the pollen fertility restoration process in CMS cotton, GO class enrichment analysis was performed for each particular DEG first (Appendix A). The majority of up-regulated genes specific to the R line of CMS-D2 had biological functions related to ‘transcription, DNA-templated’ (14 genes; 60.61%), ‘regulation of transcription, DNA-templated’ (12 genes; 51.95%), and ‘translation’ (eight genes; 34.63%). ‘Integral component of membrane (21 genes; 90.91%)’, ‘plastid’ (16 genes; 69.26%), and ‘membrane’ (14 genes; 60.61%) were the key functional terms in the cellular component category, whereas ‘transferase activity’ (11 genes; 95.24%), ‘nucleotide binding’ (10 genes; 86.58%), and ‘RNA binding’ (seven genes; 60.61%) were the most dominant terms in the molecular function category (Figure 4a, Appendix A). Additionally, most of the down-regulated genes specific to the R line of CMS-D2 had functional enrichment in the biological processes of ‘regulation of transcription, DNA-templated’ (nine genes; 90.91%), ‘transcription, DNA-templated’ (six genes; 60.61%), and ‘protein phosphorylation’ (five genes; 50.51%). At the cellular level, ‘membrane’ (eight genes; 80.81%), ‘integral component of membrane’ (six genes; 60.61%), and ‘chloroplast’ (three genes; 30.3%) had a higher portion of the genes. ‘ATP binding’ (eight genes; 80.81%), ‘transferase activity, transferring glycosyl groups’ (four genes; 40.4%), and ‘metal ion binding’ (four genes; 40.4%) were predominant terms in the molecular function category (Figure 4b, Appendix A).

According to the results of GO enrichment in the R line of CMS-D8, a higher percentage of up-regulated genes showed biological functions linked to ‘oxidation-reduction process’ (40 genes; 58.65%), ‘metabolic process’ (34 genes; 49.85%), and ‘regulation of transcription, DNA-templated’ (16 genes; 23.46%). The predominant terms included ‘integral component of membrane’ (62 genes; 90.91%), ‘membrane’ (53 genes; 77.71%), and ‘nucleus’ (nine genes; 13.2%) in the cellular component category. In contrast, ‘oxidoreductase activity’ (34 genes; 99.71%), ‘metal ion binding’ (34 genes; 99.71%), and ‘hydrolase activity’ (27 genes; 79.18%) related terms showed higher genes in the molecular function category (Figure 4c, Appendix A). Among down-regulated genes, most genes were mainly involved in the ‘oxidation-reduction process’ (49 genes; 69.6%), ‘photosynthesis’ (39 genes; 55.4%), and ‘regulation of transcription, DNA-templated’ (37 genes; 52.56%) in the biological process category. ‘Integral component of membrane’ (128 genes; 90.91%), ‘membrane’ (110 genes; 78.13%), and ‘chloroplast’ (25 genes; 17.76%) were predominant terms in the cellular component category, whereas ‘metal ion binding’ (56 genes; 79.55%), ‘ATP binding’ (39 genes; 55.4%), and ‘oxidoreductase activity’ (32 genes; 45.45%) were predominantly terms in the molecular function category (Figure 4d, Appendix A).

A Venn diagram based on GO function enrichment of specific DEGs (Figure 5a) showed significantly enriched GO analysis of DEGs in R lines (Appendix A). ‘Malate synthase activity’, ‘shade avoidance’, and ‘fruit ripening’ were the three most specific GO terms for up-regulated genes in CMS-D2. ‘Nuclear pore inner ring’, ‘protein localization to nuclear inner membrane’, and ‘hydrolase activity’ were the three most significant functions of down-regulated genes in CMS-D2. In CMS-D8, ‘metabolic process’, ‘anther wall tapetum development’, and ‘trichome morphogenesis’ were among the most significant functions for up-regulated genes, while down-regulated genes were enriched in ‘negative regulation of ethylene biosynthetic process’, ‘protein autoubiquitination’, and ‘photosystem II’ (Appendix A).

Importantly, ‘flower development’ and ‘regulation of transcription/DNA-templated’ were the common GO functions for up- and down-regulated genes in the R line of CMS-D2. ‘Oxidation-reduction process’, ‘oxidoreductase activity’, ‘heme binding’, and ‘ironionbinding’ were common GO functions for up- and down-regulated genes in the R line of CMS-D8. For common significant GO classifications, it was observed that ‘endopeptidase inhibitor activity’ was the common GO term for down-regulated genes in the R lines of both CMS-D2 and CMS-D8. On the other hand, ‘reductive pentose-phosphate cycle’, ‘ribulose-bisphosphate carboxylase activity’, and ‘photorespiration’ were the most common significant GO terms among up-regulated genes in the R line of CMS-D2 and down-regulated genes in the R line of CMS-D8. Overall, the GO enrichment analysis determined that R line genes associated with diverse biological, cellular, and molecular functions possibly transformed the sterile pollen into fertile pollen in CMS cotton (Appendix A).

### 2.4. KEGG Pathway Enrichment Analysis of Specific DEGs in Restorer Line

A KEGG pathway enrichment analysis was performed to explore the potential molecular mechanism of fertility restoration for CMS cotton. In the R line of CMS-D2, ‘RNA polymerase’, ‘glyoxylate dicarboxylate metabolism’, and ‘photosynthesis’ were the most significant pathways among specific up-regulated genes (Appendix A), while ‘taurine and hypotaurine’, ‘arachidonic acid’, and ‘thiamine metabolism’ were important annotated pathways for specific down-regulated genes (Appendix A). For the specific DEGs in the R line of CMS-D8, ‘glycosaminoglycan degradation’, ‘ABC transporters’, and ‘pentose and glucuronate interconversions’ were the most significant pathways for up-regulated genes (Appendix A). In contrast, ‘photosynthesis-antenna proteins’, ‘photosynthesis’, and ‘glycosphingolipid biosynthesis-ganglio series’ were the most significant pathways under down-regulated genes (Appendix A).

Interestingly, the Venn diagram based on the KEGG pathway enrichment analysis of specific DEGs (Figure 5b) revealed that RNA polymerase, protein processing in the endoplasmic reticulum, pyrimidine, and purine metabolism pathways were the most significant for up-regulated genes in the R line of CMS-D2 (Appendix A). For down-regulated genes, the most enriched pathways were taurine, hypotaurine, and cyanoamino acid metabolism. The majority of up-regulated genes in the R line of CMS-D8 were involved in ABC transporters, glycosaminoglycan degradation, pentose and glucuronate interconversions, and plant-pathogen interaction pathways. However, photosynthesis—antenna proteins, nitrogen metabolism, diterpenoid biosynthesis, biosynthesis of unsaturated fatty acids, carbon fixation in photosynthetic organisms, and alpha-linolenic acid metabolism—were the most enriched pathways for down-regulated genes (Appendix A). Further analysis showed that photosynthesis and glyoxylate and dicarboxylate metabolism were common KEGG pathways among up-regulated genes of CMS-D2 and down-regulated genes of CMS-D8 (Appendix A). Importantly, the pathway enrichment analysis determined that the combined contribution of various metabolic pathways plays a key role in inducing pollen fertility in the R line of CMS cotton.

### 2.5. Co-Localization Analysis of the Up-Regulated DEGs in Two R Lines Located on Chr_D05 and Identification of Candidate Restoration Genes

Our previous studies identified that the target region of restorer gene *Rf_1_* is at 37.12 Mb--54.29 Mb of chromosome D05, and *Rf_2_* is located at 54.30 Mb--55.78 Mb of chromosome D05 [34,39]. This research therefore further physically mapped the up-regulated DEGs of restorer lines on Chr_D05 to identify restorer candidate genes. Importantly, 28 of the 204 up-regulated genes in CMS-D2 were mapped on Chr_D05, 81 of the 557 up-regulated genes in CMS-D8 were mapped on Chr_D05, and seven genes that overlapped among both systems were mapped on Chr_D05 (Appendix A). The physical locations of 102 up-regulated genes in both CMS-D2 and CMS-D8 R lines, as well as the target regions of *Rf_1_* and *Rf_2_* on Chr_D05, are shown in Figure 6. The target region of *Rf_1_* is the green column, whereas the target region of *Rf_2_* is the purple column. In addition, 19 of 21 specifically up-regulated genes of CMS-D2 are in green (except 2 novel genes), 71 of 74 specifically up-regulated genes of CMS-D8 are in blue (except 3 novel genes), and seven genes co-up-regulated in both CMS-D2 and CMS-D8 are in bold red. Importantly, two (*Ghir_D05G032210* and *Ghir_D05G032450*) out of seven genes co-up-regulated in both R lines were located in the predicted target region of *Rf_1_*, while one gene, *Ghir_D05G035290*, specifically up-regulated in R lines of CMS-D8, was located in the predicted target region of *Rf_2_* (Figure 6). Interestingly, heat-map analysis revealed that overlapped seven genes, including *Ghir_D05G032210*, *Ghir_D05G036230*, *Ghir_D05G036240*, *Ghir_D05G032450*, *Ghir_D05G035690*, *Ghir_D05G035720*, and *Ghir_D05G036250*, had significant differential expression in R lines compared to A and B lines (Figure 7). These overlapped genes among R lines of both CMS systems could be important to understanding the genetic mechanism of fertility restoration in cotton. Their expression is therefore further validated using real-time qRT-PCR. The results showed these seven DEGs were specifically highly expressed in R lines of both CMS systems, which had similar expression trends in qRT-PCR and RNA-seq data (Figure 8). In brief, we speculated that up-regulated DEGs located in target regions of *Rf_1_* and *Rf_2_* could be candidate fertility restorer genes for both CMS systems. However, further functional validation research should be critical to exploring how these genes participate in the process of fertility restoration in cotton.

## 3. Discussion

### 3.1. Overview of Differential Transcriptome Dynamics in Two Cotton CMS Systems

The CMS system is an important tool for commercial hybrid seed production in many agronomic crops, such as rice, maize, wheat, and cotton. Currently, the two main well-developed CMS systems in cotton are CMS-D2 and CMS-D8 [4]. The limited available knowledge on the fertility restoration process, however, further reduced the wider application of both CMS systems in cotton. A better understanding of the genetic mechanisms of fertile pollen could enhance the efficiency of future cotton breeding programs based on CMS systems.

To my particular interest, transcriptomes have been widely used to investigate molecular mechanisms and mine-related functional genes for pollen fertility in recent years. For example, Guan et al. [42] explored the molecular mechanism of the triploidy effect in the pearl oyster *Pinctada fucata* by differential transcriptomics of the diploid and triploid pearl oyster *Pinctada fucata*. Hamid et al. [43] found that *DYT1* and *AMS* were aberrantly expressed in pollen developmental pathways of CGMS in cotton through transcriptome analysis. Our previous studies have also characterized the full-length transcripts in the anthers of both CMS-D2 and CMS-D8 systems [40,41]. In addition, *Gh_D05G3427* has been identified as a potential candidate restorer gene for CMS-D2 cotton by transcriptome analysis [27]. However, the transcriptional differences between the two CMS systems are still largely unclear. Here, we comprehensively compared the long-read transcript data of the CMS-D2 and CMS-D8 systems in cotton. The results showed that a large number of DEGs were identified in the R lines of both CMS systems compared to the respective A and B lines. Moreover, some genes with specific differential regulation were also found to exist in R lines (Figure 1 and Figure 2). These results enabled us to understand that transcriptional reprogramming in R lines might contribute to the retrieval of pollen fertility for CMS cotton. Consistent with the results of our previous studies [27,34,44], most of the specific DEGs in R lines were distributed on Chr_D05 of upland cotton in both CMS-D2 and CMS-D8 systems (Figure 2 and Figure 3), suggesting that the expression profiles of these genes may possibly interact with the restorer genes *Rf_1_* or *Rf_2_* to recover the pollen fertility in cotton. Thus, these genes can be part of the gene regulatory network linked to fertility restoration in CMS cotton.

### 3.2. Differential Gene Regulation Network Probably Influences the Retrieval of Pollen Fertility for CMS-D2 and CMS-D8 Cotton

Functional annotations provide mechanistic insight into the regulatory network of target traits. In this study, DEGs specific to R lines were found to be involved in several biological, cellular, and molecular functions (Figure 4 and Figure 5). In the restorer line of CMS-D8, the differential regulation of genes involved in anther wall tapetum development and enriched in the biosynthesis of unsaturated fatty acids may be responsible for normal pollen development. Previously, it has been reported that the differentiation and functional activities of the anther tapetum cell as a vegetative organ mediate pollen development in flowering plants, and abnormal growth of tapetum cells leads to pollen sterility [45]. In plants, the unsaturated fatty acids not only act as a reserve of energy but also defend plants from several stresses. Apart from other responses, their defense is characterized by signaling related to oxidative burst and programmed cell death, often occurring in male sterile lines [46]. The specifically expressed genes in R lines may facilitate the balance between the biosynthesis of unsaturated fatty acids and oxidative stress in developing pollen. The imbalance in this pathway most probably produced male sterility in the A line of CMS-D8 cotton [29]. Recent research has shown disruption in unsaturated fatty acid metabolism ultimately led to the decline of pollen fertility in the CMS-D2 cotton restorer line [47]. Interestingly, the differential genes specific to the restorer line of CMS-D2 were enriched for functions associated with flower development, and most of these genes showed pathway annotations with the biosynthesis of various secondary metabolites. In higher plants, flower development is a complex process regulated by a dynamic network of genes related to floral organ identity [48]. Additionally, recent studies predicted the essential role of secondary metabolites to ensure male fertility in crops [49,50]. In this way, the combined effects of flower-encoding genes and secondary metabolites might regulate various aspects linked to the retrieval of male fertility in CMS-D2 cotton. Furthermore, our results further determined that genes involved in photosynthesis and dicarboxylate metabolism were the only pathways that overlapped among both CMS systems. These key pathways could therefore facilitate male fertility restoration in CMS-D2 as well as in CMS-D8 cotton. The difference in the cytoplasm of both lines could be the primary reason that a different fertility restoration mechanism may exist in CMS-D2 and CMS-D8 cotton. In addition, the genetic effects of the fertility restorer gene *Rf_2_* specific to CMS-D8 may lead to a different fertility revival mechanism compared with CMS-D2 cotton. However, further functional validation research should be carried out to explore the difference in the genetic basis of fertility restoration for both CMS systems in cotton.

### 3.3. Discovery of Gene Regulatory Network Located in the Target Region of Fertility Restorer Genes in CMS Cotton

Due to the substantial contribution of CMS-based hybrid breeding, the identification and gene mapping of fertility restorer (*Rf*) genes have been extensively performed in several agronomic crops. In the case of cotton, the development of molecular markers linked with the *Rf_1_* and *Rf_2_* restorer genes has progressed in recent years [34,35,38,51,52]. An earlier study showed that *Rf_1_* and *Rf_2_* are not allelic genes but tightly linked on chromosome D05 of upland cotton [53]. Molecular evidence further identified that the *Rf_1_* gene was limited to a 100-kb region [32], and Wu et al. [34] mapped the *Rf_1_* gene in a region of 0.279 cM on chromosome D05. In addition, Feng et al. [39] utilized InDel markers to locate the *Rf_2_* gene in a 1.48 Mb region on chromosome D05. Combined with our previous mapping of *Rf_1_* and *Rf_2_*, this study further identified key genes specifically expressed in the R lines and located in the target region of fertility restorer genes (Figure 6). In particular, our study analyzed seven key genes that showed specific up-regulation in the restorer lines of both CMS-D2 and CMS-D8 cotton (Figure 7). Remarkably, two genes denoted as *Ghir_D05G032210* and *Ghir_D05G032450* were identified in the predicted target region of *Rf_1_* on Chr_D05. These genes may be related to *Rf_1_* and participate in the process of fertility restoration for CMS-D2 as well as CMS-D8. *Ghir_D05G035690* was mapped in the predicted target region of *Rf_2_* on Chr_D05 (Figure 6). This gene might interact with the restorer gene *Rf_2_* and mediate male gametophyte development in CMS-D8 cotton. Among the three main genes, *Ghir_D05G032450* encodes ADP-ribosylation factor 1 (*ARF1*). Previous studies have demonstrated that *Arf1* is expressed predominantly in the reproductive organs of cotton and probably influences the differentiation and development of reproductive organs [54,55,56]. Consistent with previous results, we found that the expression of the *Ghir_D05G032450/ARF1* gene was up-regulated in the restorer lines of both CMS-D2 and CMS-D8, and it was identified in the target region of *Rf_1_* on Chr_D05 (Figure 6, Figure 7 and Figure 8). Due to the fact that *Rf_1_* can retrieve pollen fertility in both CMS systems [4], it is most probably a candidate gene linked with the complex gene regulatory network of fertility restoration for both CMS cottons. Comparatively, *Ghir_D05G035690* could be a candidate gene with the ability to restore pollen fertility only for CMS-D8 cotton. Concisely, this study identified an important gene regulatory network tightly linked with the revival of pollen fertility for both CMS systems in cotton. However, further research at the gene functional level is required to understand how these genes differentially facilitate fertility restoration associated with *Rf_1_* or *Rf_2_*.

## 4. Materials and Methods

### 4.1. Plant Materials

The plant materials analyzed in this study included A, B, and R lines of both CMS-D2 and CMS-D8 cotton systems. The CMS-D2 system was comprised of the A line with *G. harknessii* D_2-2_ cytoplasm ([S(*rf_1_rf_1_*)]), the B line with upland cotton cytoplasm ([N(*rf_1_rf_1_*)]), and the R line with a homozygous fertility dominant allele ([S(*Rf_1_Rf_1_*)]). Similarly, the CMS-D8 system was comprised of the A line with *G. trilobum* D_8_ cytoplasm ([S(*rf_2_rf_2_*)]), the B line with upland cotton cytoplasm (N(*rf_2_rf_2_*)), and the R line with a homozygous fertility dominant allele ([S(*Rf_2_Rf_2_*)]). All materials were planted at the experimental farm of the Institute of Cotton Research of the Chinese Academy of Agricultural Science (ICR-CAAS), Anyang, Henan, China. The anthers from flower buds with lengths of 1, 2, 3, and 4 mm were harvested centrally from at least 50 plants, pooled in three biological replications for each line of both CMS systems, immediately frozen in liquid nitrogen, and finally stored at −80 °C in a freezer before further use.

### 4.2. RNA Extraction, Sequencing Library Construction, and Data Analysis

The Sigma Spectrum Plant Total RNA Kit (Sigma-Aldrich, Saint Louis, MO, USA) was used to extract high-quality total RNA for each sample following the manufacturer’s procedure. The brief details on RNA-Seq library construction, PacBio sequencing, and workflow of quality data analysis can be seen for CMS-D2 and CMS-D8 in our previously published research results [40,41].

### 4.3. Differential Expression and GO and KEGG (Kyoto Encyclopedia of Genes and Genomes) Enrichment Analysis

Here, the high-quality sequenced reads were blasted against the reference genome of *G. hirsutum* TM-1 to calculate the abundance of gene expression. DESeq was used to analyze and screen biological repeatable samples and non-biological repeatable samples. A false discovery rate (FDR) < 0.05 and a fold change > 1 were the standard criteria to screen differentially expressed genes (DEGs) between target groups. The OmicStudio tool (URL https://www.omstudio.cn/tool, accessed on 30 June 2022) was used to retrieve GO functional classification and pathway enrichment analysis for DEGs specific to restorer lines of both CMS systems.

### 4.4. Chromosome Location of DEGs on D05 in the Target Region of Restorer Genes

The target mapping region of restorer genes *Rf_1_* and *Rf_2_* on chromosome D05 was previously identified by our research team [34,39]. Here, specific DEGs in restorer lines located in the target regions of fertility restorer genes were identified as the candidate *Rf_1_* and *Rf_2_* genes. The final physical map was drawn using the MapChart v2.3 software [57].

### 4.5. qRT-PCR Validation for DEGs

Total RNA was extracted from the anthers of each sample using the TIANGEN RNAprep Pure Plant Plus Kit (polysaccharides and polyphenolics-rich; DP441) according to the manufacturer’s procedure, and DNA contamination was also removed with the DNase I contained in the Kit. Reverse transcription reactions and qRT-PCR of DEGs were performed using the PrimerScript™ RT Master Mix for Perfect Real Time (RR036A, TaKaRa, Kyoto, Japan) and TransStart^®^ Top Green qPCR SuperMix (AQ131, TransGen Biotech, Beijing, China), respectively, according to the manufacturer’s instructions. The housekeeping gene upland cotton *His3* (histone 3) was used as an internal control for data normalization, and the relative gene expression levels were calculated with the 2^−△△CT^ method [58]. Each gene in each sample was analyzed with three biological replicates and two technical repetitions to ensure the reliability of the results. Gene-specific primers for qRT-PCR were designed using Oligo7 software and [59] synthesized commercially (BioSune Biotechnology, Shanghai, China), and are shown in Appendix A.

## 5. Conclusions

In this study, we systemically compared the transcriptional differences between the CMS-D2 and CMS-D8 systems in cotton for the first time. The comprehensive comparative analysis determined a significant differential gene regulation in the restorer lines compared to the sterile and maintainer lines in both CMS systems. In addition, a substantial portion of specific DEGs in the restorer lines of both CMS systems were also identified. GO and KEGG enrichment analysis revealed that these specific DEGs could be involved in regulating anther development or pollen fertility restoration for two types of CMS cotton through various functional activities and biological processes. Further analysis identified potential candidate fertility restorer genes in the mapping region of *Rf_1_* or *Rf_2_*. The results obtained in this study will help to further explore the precise genetic relationship between restorer genes *Rf_1_* and *Rf_2_* and also open a new door for dissecting differences in the genetic determinants of pollen fertility restoration for CMS-D2 and CMS-D8 cotton.

## Figures and Tables

**Figure 1 ijms-24-10759-f001:**
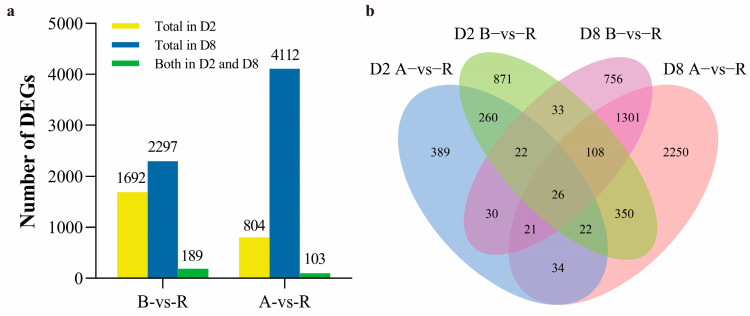
Summary of the differentially expressed genes in CMS-D2 and CMS-D8. (**a**) Number of total and shared DEGs in the two comparisons. (**b**) Venn diagrams showing the distribution of DEGs among the two comparisons.

**Figure 2 ijms-24-10759-f002:**
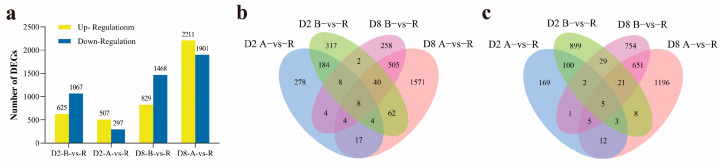
Summary of up-regulated and down-regulated in CMS-D2 and CMS-D8. (**a**) Number of up-regulated and down-regulated DEGs in the two comparisons. (**b**) Venn diagrams showing the distribution of up-regulated DEGs among the two comparisons. (**c**) Venn diagrams showing the distribution of down-regulated DEGs among the two comparisons.

**Figure 3 ijms-24-10759-f003:**
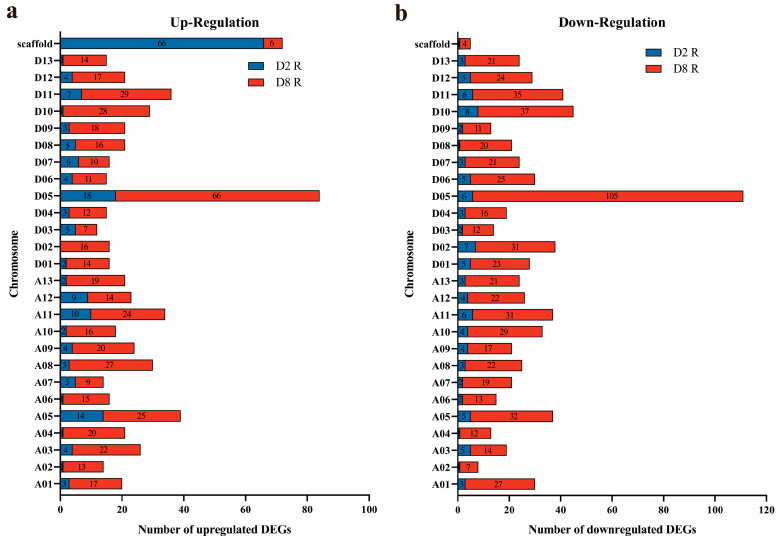
Number of DEGs specifically expressed in the R line on different chromosomes. (**a**) Number of up-regulated genes on different chromosomes. (**b**) Number of down-regulated genes on different chromosomes. The Y-axis represents different chromosomes. The X-axis and numbers on each bar represent the DEG numbers on each chromosome.

**Figure 4 ijms-24-10759-f004:**
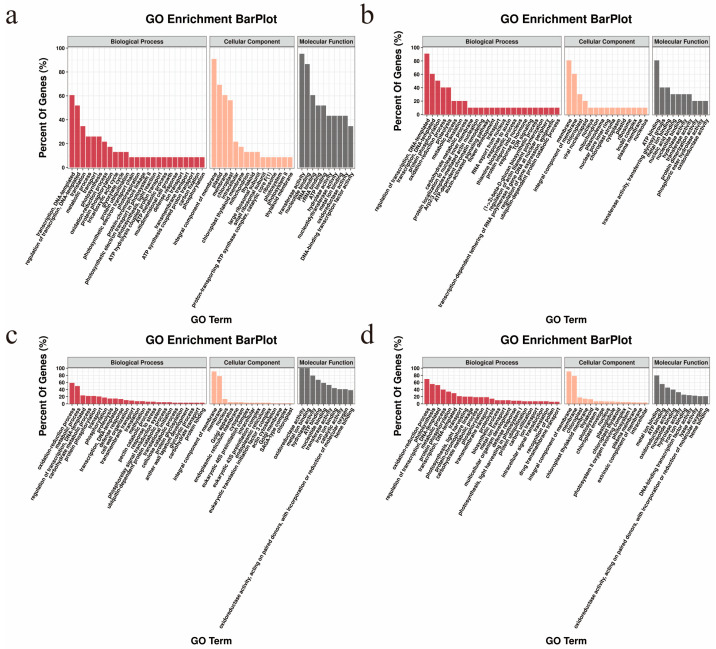
Gene ontology analysis of specifically expressed genes in CMS-D2 and CMS-D8. (**a**,**b**) Functional enrichment among the DEGs specifically up-regulated and down-regulated in CMS-D2 R line. (**c**,**d**) Functional enrichment among the DEGs specifically up-regulated and down-regulated in CMS-D8 R line. The names of the GO categories are listed along the x-axis. The degree of GO enrichment is represented by the −log10 (*p*-value).

**Figure 5 ijms-24-10759-f005:**
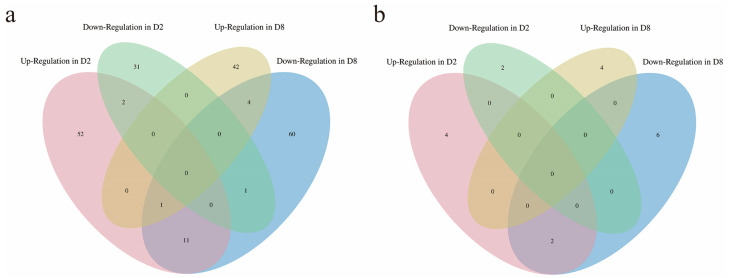
GO and KEGG significantly enriched for DEGs in R lines. (**a**) Venn diagram showing the distribution of GO functional classifications. (**b**) Venn diagram showing the distribution of the KEGG pathways.

**Figure 6 ijms-24-10759-f006:**
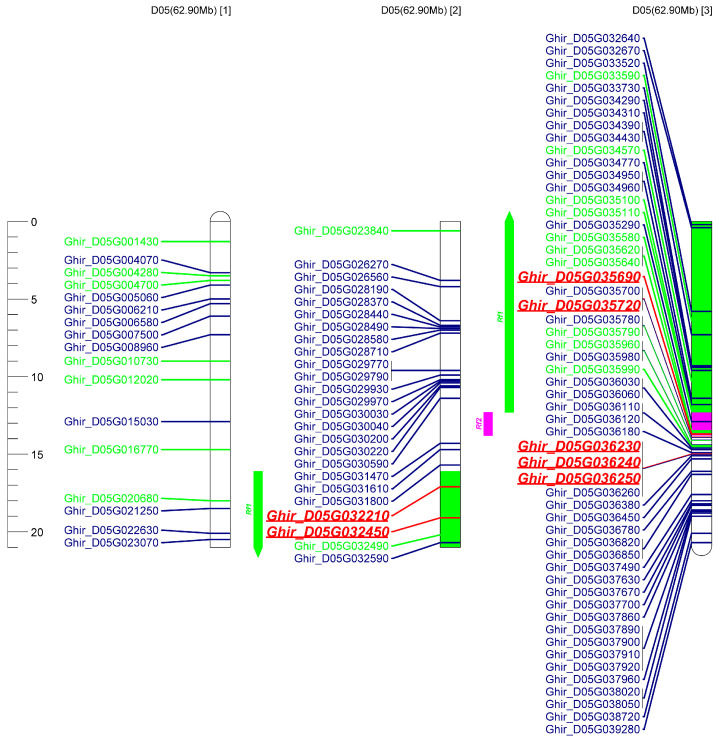
Physical location distribution of up-regulated DEGs on Chr_D05 from CMS-D2 and CMS-D8. The green column represents the *Rf_1_* target region; the purple column represents the *Rf_2_* target region; the genes in green represent up-regulated DEGs in the CMS-D2 R line. The genes in blue represent up-regulated DEGs from CMS-D8; the genes in red and bold represent co-upregulated DEGs from CMS-D2 and CMS-D8. The 62.90 Mb represents the chromosome length; the numbers 1, 2, and 3 in bracket indicate the first, second, and third segments of Chr_D05.

**Figure 7 ijms-24-10759-f007:**
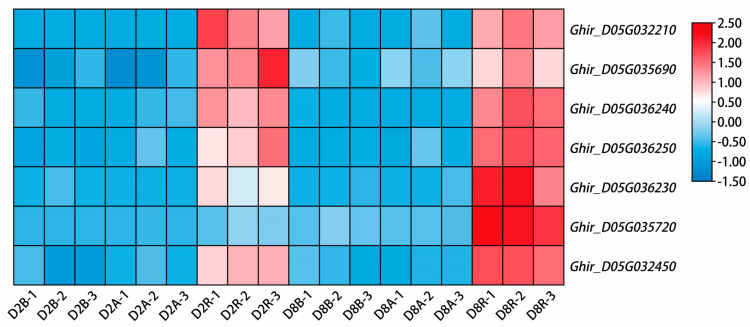
Heat map of FPKM values of co-expressed DEGs in the restorer lines of CMS-D2 and CMS-D8. High (red) or low (blue) expression levels were displayed based on normalized data (color bar on the right side of the heat map) generated using an integrative toolkit TBtools. A: sterile line, B: maintainer line, R: restorer line. D2: CMS-D2, D8: CMS-D8.1 2 3: three repetitions.

**Figure 8 ijms-24-10759-f008:**
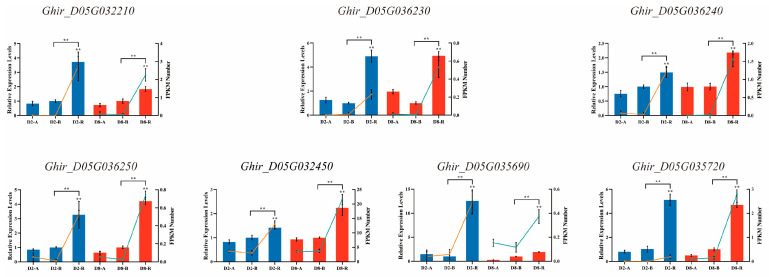
qRT-PCR analysis of gene expression compared with the RNA-seq data. The blue columns represent the relative expression levels of the genes in CMS-D2 and the red columns represent the relative expression levels of the genes in CMS-D8; the orange solid lines represent the RNA-seq reads in CMS-D2 and the green solid lines represent the RNA-seq reads in CMS-D8. T-test was two-tailed. ** represents significant differences from the control at *p* < 0.01, as determined by the two-tailed *t*-test. A: sterile line, B: maintainer line, R: restorer line. Blue, CMS-D2. Red, CMS-D8.

## Data Availability

Data supporting report results can be found in the Appendix A.

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
