# Peer review of "Comparative Transcriptome Profiling of CMS-D2 and CMS-D8 Systems Characterizes Fertility Restoration Genes Network in Upland Cotton"

_ijms, 2023, doi:10.3390/ijms241310759_

Round 1

Reviewer 1 Report

The present study compared the transcriptional datasets of CMS-D2 and CMS-D8 lines by using full-length transcripts sequencing. Further analyzed gene expression differences, identified fertility-related genes, mapped key genes into target regions of Rf1/Rf2, and confirmed results accuracy with qRT-PCR. The results from this study will be useful to explore the precise genetic relationship between Rf1 and Rf2. This study reveals useful insights into the regulatory mechanism that retrieve pollen fertility of male sterile lines of CMS cotton and further provides an important foundation to understand the genetic architecture of pollen fertility restoration of CMS cotton. The manuscript is well-structured and well-discussed. However, some points should be checked and corrected before it's accepted in this journal. 

Therefore, according to my comments, I recommended the publication of the paper after minor revision.

[1]   Please speculate on the results. The discussion must improve.

[2]   In conclusion, The authors should add the significance of this research and its potential practical application.

[3]   The MS English needs to be improved. The article's English must be carefully checked for grammatical errors.

Reviewer 2 Report

Dear authors.

The manuscript is well-written. However, there are some minor issues that can improve the quality of the manuscript. My suggestions are included in the attached PDF file. 

Sincerely.

Author Response

Thank you very much for your positive and constructive suggestions to improve our manuscript. We have made correction according to your comments. You can view the modified article by the revised version.

Reviewer 3 Report

All works related to research with the most important agricultural crops are relevant.

there are some minor comments

70- rephrase the sentence. "genes related to plants". The manuscript is generally devoted to the cotton plant.

109 indicate that A, B and R are different lines of cotton

figure 6 shows the numbers DO5 ( ....)[1,2,3], but there is no explanation in the figure caption

166-xis?

321 and beyond - when authors are given, it is better to indicate the reference immediately

The manuscript is neatly formatted, except for references. All references must be made in accordance with the rules of the journal

specify references 12,13,14,15

45 - which edition?

52.54 - only the year is indicated
